



# A parameterization of Long-Continuing-Current (LCC) lightning in the lightning submodel LNOX (version 3.0) of the Modular Earth Submodel System (MESSy, version 2.54)

Francisco J. Pérez-Invernón[1], Heidi Huntrieser[1], Patrick Jöckel[1], and Francisco J. Gordillo-Vázquez[2]

[1]Deutsches Zentrum für Luft- und Raumfahrt, Institut für Physik der Atmosphäre, Oberpfaffenhofen, Germany
[2]Instituto de Astrofísica de Andalucía, CSIC, Glorieta de la Astronomía s/n, Granada, Spain

**Correspondence:** Francisco J. Pérez-Invernón (FranciscoJavier.Perez-Invernon@dlr.de, https://orcid.org/0000-0003-2905-3854)

**Abstract.** Lightning flashes can produce a discharge in which a continuing electrical current flows for more than 40 ms. This type of flashes are proposed to be the main precursors of lightning-ignited wildfires and also to trigger sprite discharges in the mesosphere. However, lightning parameterizations implemented in global atmospheric models do not include information about the continuing electrical current of flashes. The continuing current of lightning flashes cannot be detected by conven-

5 tional lightning location systems. Instead, these so-called Long-Continuing-Current (LCC) flashes are commonly observed by Extreme Low Frequency (ELF) sensors and by optical instruments located in space. Previous reports of LCC lightning flashes tend to occur in winter and oceanic thunderstorms, which suggests a connection between weak convection and the occurrence of this type of discharge.

In this study, we develop a parameterization of LCC lightning flashes based on a climatology derived from optical lightning

10 measurements reported by the Lightning Imaging Sensor (LIS) on-board the International Space Station (ISS) between March 2017 and March 2020. We use meteorological data from reanalyses to find a global parameterization that uses the vertical velocity at 450 hPa pressure level as a proxy for the ratio of LCC to typical lightning in thunderstorms. We implement this parameterization into the LNOX submodel of the Modular Earth Submodel System (MESSy) for usage within the EMAC model, and compare the observed and the simulated climatologies of LCC lightning flashes using six different lightning pa-

15 rameterizations. We find that the best agreement between the simulated and the observed spatial distribution is obtained when using a novel combined lightning parameterization based on the cloud top height over land and on the convective precipitation over ocean.



# 1 Introduction

Lightning flashes are formed by electrical discharges with duration ranging between a few hundred of microseconds and hundreds of milliseconds (Rakov and Uman, 2003). Lightning flashes containing a discharge in which a continuing electrical current flows during more than 40 ms are usually referred to as Long-Continuing-Current lightning (LCC-lightning) (Brook et al., 1962). LCC-lightning has been associated with lightning-ignited fires (e.g., Fuquay et al., 1967; Latham and Williams, 2001; Pineda et al., 2014; Pérez-Invernón et al., 2021b), as the long duration of the discharge can favor ignition. This assumption is supported by laboratory experiments (e.g., McEachron and Hagenguth, 1942; Feng et al., 2019; Zhang et al., 2021). LCC-lightning is also proposed to be the main precursor of sprites (Bell et al., 1998; Cummer and Füllekrug, 2001; Cummer, 2003), a type of Transient Luminous Event (TLE) taking place in the mesosphere above thunderclouds (Pasko et al., 2012; Gordillo-Vázquez and Pérez-Invernón, 2021). Despite the evidences of the role of LCC-lightning in lightning-ignited fires and in the production of sprites, there are still noteworthy uncertainties in the relationship between the meteorological conditions of thunderstorms and the occurrence of LCC-lightning.

Lightning Location Systems (LLS) provide global and continuous monitoring of the lightning activity all around the world. LSS are mostly assembled by Very Low Frequency (VLF) sensors that can detect the electromagnetic signature emitted by lightning flashes (Nag et al., 2015). VLF sensors are sensible to the far field component of the electromagnetic field produced by lightning, that is proportional to the peak current and decreases with distance following an inverse-square law. However, the continuing phase of LCC-lightning may lack of a high peak current (Pineda et al., 2014). The continuing phase of LCC-lightning produces an electrostatic field (also called near field) that decreases with the distance following an inverse-cubic law. Therefore, LLS are not useful to provide information about the continuing phase of lightning.

Fuquay et al. (1967) and Adachi et al. (2009) showed that the optical signal emitted by lightning discharges can be related to the duration of the electrical discharge. Bitzer (2017) investigated for the first time the tropical and mid-latitude climatology of LCC-lightning discharges from optical lightning measurements reported by the Lightning Imaging Sensor (LIS) on-board the Tropical Rainfall Measuring Mission (TRMM) satellite following a low-Earth orbit between 1997 and 2015 and providing lightning measurements in the range of latitudes between 35°N and 35°S. Based on the observation that LCC-lightning discharges tend to occur in oceanic and winter thunderstorms, Bitzer (2017) proposed that thunderstorms with weaker updrafts would produce small charging rates, allowing the charging process to develop larger charge regions before the onset of lightning and providing the discharge with more energy to be transferred.

Lightning and TLEs are sub-grid phenomena that cannot be self-consistently implemented in global atmospheric models. However, the process of separation of electrical charges that produces lightning is highly influenced by dynamic and thermodynamic processes (Showalter, 1953). Therefore, lightning and TLE activity are parameterized in global atmospheric models using meteorological variables as proxies (e.g., Tost et al., 2007; Murray et al., 2012; Pérez-Invernón et al., 2019; Gordillo-Vázquez et al., 2019). In the same way, relating the occurrence of LCC-lightning activity to large scale meteorological parameters could be helpful to improve the parameterization of lightning-ignited fires in global climate models and to implement the occurrence of sprites. In this study, we present a simple LCC-lightning parameterization which relates the ratio





of LCC-lightning to typical lightning in thunderstorms with the updraft strength at a specific altitude. We implement this novel parameterization as an upgrade of the `LNOX` submodel (Tost et al., 2007) of the Modular Earth Submodel System (MESSy) and test it with the European Center HAMburg general circulation model (ECHAM) / MESSy Atmospheric Chemistry (EMAC) model (v2.54). We test the parameterization by comparing the simulated seasonal spatial distribution of LCC-lightning during 2018 with lightning data reported by LIS onboard the International Space Station (ISS-LIS).

## 2 Observations

### 2.1 Lightning measurements and LCC-lightning

The climatology of LCC-lightning flashes developed by Bitzer (2017) was based on TRMM-LIS data acquired during a 12 year period (2002 –2013) and in the range of latitudes between 35°N and 35°S (Christian et al., 2003; Cecil et al., 2014). Bitzer (2017) proposed a method to identify LCC-lightning flashes from the groups reported by TRMM-LIS. ISS-LIS detects optical emissions from lightning with a frame integration time of 1.79 ms (Bitzer and Christian, 2015) and with a spatial resolution of 4 km (Blakeslee et al., 2020), while the spatial resolution of TRMM-LIS was approximately 5 km (Christian et al., 2003; Cecil et al., 2014). LIS assorts contiguous events into groups, and clusters groups into flashes with a temporal criteria of 330 ms and a spatial criteria of 5.5 km (Mach et al., 2007). Bitzer (2017) proposed that optical emissions *detected in five or more consecutive frames* (time contiguous groups), that are in the same flash, can be classified as a LCC(>9 ms) lightning flash. In the same manner, optical emissions *detected in ten or more consecutive frames* (time contiguous groups), that are in the same flash, can be classified as a LCC(>18 ms) lightning flash. LIS detects all sort of lightning, that is, intra-cloud (IC) and cloud-to-ground (CG).

The operations of TRMM-LIS ended in 2015, followed by the placement of a similar instrument onboard the ISS for a 4 years mission starting in March 2017 covering latitudes between 54.3°N and 54.3°S (Blakeslee et al., 2020). We use the method proposed by Bitzer (2017) to produce global climatologies of LCC(>9 ms) and LCC(>18 ms) lightning flashes based on ISS-LIS lightning measurements between March 2017 and March 2020.

We show in Fig. 1 the obtained total annual lightning flash density (IC + CG lightning), LCC(>9 ms)-lightning flash density, and ratios between LCC(>9 ms)- and LCC(>18 ms)-lightning to typical total lightning. The first panel, showing the total annual lightning flash density composed by a total number of $3.5 \times 10^6$ flashes, is in agreement with the ISS-LIS annual gridded climatology reported by Blakeslee et al. (2020). The peak flash density occurs over central Africa, while there are other regions with significant lightning activity, such as the Himalayas and India, some regions of Paraguay, Argentina, Brazil, Venezuela, North America and the Maritime Continent. The contrast between land and ocean is also shown.

The second panel of Fig. 1 shows the obtained LCC(>9 ms)-lightning flash density. In total, we have obtained 234007 LCC(>9 ms)-lightning flashes. Areas with high lightning activity are also regions with a high abundance of LCC(>9 ms)-lightning flashes. However, other regions are also LCC(>9 ms)-lightning hotspots. This feature of LCC(>9 ms)-lightning flash spatial distribution is clearer shown in the third panel of Fig. 1, showing the ratio of LCC(>9 ms)-lightning to all lightning reported by ISS-LIS in each cell. The ratio of LCC(>9 ms)-lightning to all lightning is higher over ocean than over land.





Over land, the ratio of LCC(>9 ms)-lightning to all lightning reaches its maximum over Australia, Southern Chile, Canada and Eastern Europe. There are also some small areas with a high ratio of LCC(>9 ms)-lightning to all lightning in Southern Africa, China, Japan and in the Western coast of North America. The obtained spatial distribution of the ratio of LCC(>9 ms)-lightning to all lightning is in agreement with Fig. 6 of Bitzer (2017) between 35°N and 35°S latitude. Interestingly, a high ratio downwind of North America, Argentina, South Africa and Australia can be seen. All these regions are well-known regions for intercontinental transport of trace gases (start of warm conveyor belts) (Eckhardt et al., 2004). The ratio is also high in the outflow from West Africa to South America, a transport route for dust and biomass burning.

The fourth panel of Fig. 1 shows that the spatial distribution of the ratio of LCC(>18 ms)-lightning to all lightning flashes is nearly similar to the spatial distribution of the ratio of LCC(>9 ms)-lightning to all lightning flashes, as both ratios are higher over ocean than over land, and show maxima over the same continental areas. We obtain a total of $2.6 \times 10^4$ LCC(>18 ms)-lightning flashes. This quantity is significantly lower than the obtained total number of lightning flashes and LCC(>9 ms)-lightning flashes ($3.5 \times 10^6$ and $2.3 \times 10^5$, respectively). Therefore, the spatial distribution of the ratio of LCC(>18 ms)-lightning to all lightning flashes is less smooth than the spatial distributions of the ratio of LCC(>9 ms)- to typical-lightning flashes.

## 2.2 Meteorological data

Thunderstorm electrification processes are highly influenced by meteorological conditions producing the rising of moist air reaching the level of free convection below the 500 hPa level (Showalter, 1953). Several of the most used lightning parameterizations are based on meteorological variables at the 440 hPa pressure level that are related with convection. For example, the parameterizations by Allen and Pickering (2002) and Finney et al. (2014) use the updraft strength at 440 hPa pressure level and the cloud ice flux at 440 hPa to estimate the lightning activity, respectively. The 440 hPa pressure level is typically chosen to parameterize lightning because temperature is about -25°, favoring supersaturation and the co-existence of a mixture of ice particles and liquid droplets (Korolev and Mazin, 2003) that contributes to electrification (Khain et al., 2012). Other lightning parameterizations employ some meteorological variables that are also related with convection, such as the parameterization by Grewe et al. (2001), that uses the updraft velocity in clouds as a proxy for lightning activity, or the parameterizations by Price and Rind (1992) and Luhar et al. (2021), that use the Cloud Top Height (CTH).

As proposed by Bitzer (2017), updrafts could play a role for the production of LCC-lightning. Therefore, we propose using the vertical velocity at 450 hPa pressure level as a proxy for LCC-lightning activity. The 450 hPa pressure level has been chosen because it is the nearest level to the 440 hPa level in the ERA5 grid where ice particles and liquid droplets can co-exist. We use 1-hourly ERA5 vertical velocity at 450 hPa pressure level with a horizontal resolution of 0.25° provided by the European Centre for Medium-Range Weather Forecasts (ECMWF) ERA5-reanalysis data set (Hersbach et al., 2020). In particular, we extract 1-hourly global values of the vertical velocity at 450 hPa for the entire period between March 1, 2017 and February 28, 2018.





## 3 LCC-lightning model description

In this section, we describe the developed LCC-lightning parameterization. We start with a brief description of EMAC and
the `LNOX` submodel in section 3.1. In section 3.2, we introduce the LCC-lightning parameterization. Finally, we describe the
implementation of the developed LCC-lightning parameterization in the `LNOX` submodel in section 3.3.

### 3.1   Chemistry–climate model EMAC and `LNOX` submodel

The developed LCC-lightning parameterization is implemented as a modification of the `LNOX` submodel of the Modular Earth
Submodel (MESSy v2.54) and tested with the EMAC model. The EMAC model is a numerical chemistry-climate model that
couples the fifth generation European Center HAMburg general circulation model (ECHAM5; (Roeckner et al., 2006)) and
the second version of Modular Earth Submodel System (MESSy) to link multi-institutional computer codes, known as MESSy
submodels (Jöckel et al., 2010, 2016) . Such submodels are used to describe tropospheric and middle atmosphere processes
and their interaction with oceans, land, and influences coming from anthropogenic emissions.

The `LNOX` submodel estimates the flash density and production of $NO_x$ by lightning by using different lightning parame-
terizations (Tost et al., 2007) and a scaling factor that ensures a global lightning occurrence rate of ∼45 flashes per second
(Christian et al., 2003; Cecil et al., 2014). For the present study, we use the parameterization of Grewe et al. (2001) based on
the updraft velocity in clouds (here referred as $G_{updr}$), the parameterization based on the CTH by Price and Rind (1992, here
referred as $P_{cth}$), the two parameterizations by Allen and Pickering (2002) based on the convective precipitation (here referred
as $A_{prec}$) and the updraft strength at the 440 hPa pressure level (here referred as $A_{updr}$), respectively, and a novel combination
of the parameterization based on the CTH (Price and Rind, 1992) for land and on the updraft strength at 440 hPa pressure level
(Allen and Pickering, 2002) for ocean (here referred as $P_{cth} + A_{prec}$) suggested by us. In addition, we have implemented an
extra lightning parameterization based on the improved cloud-height-based parameterization reported by Luhar et al. (2021,
eq. (17-18), here referred as $L_{cth}$). The lightning parameterizations used in this study are summarized in Table 1.

### 3.2   Parameterization of LCC-lightning based on the updraft strength

In this section, we investigate the relationship between the ratios of LCC(>9 ms)- and LCC(>18 ms)-lightning to typical
lightning flashes and the updraft strength from ERA5 reanalysis.

First, we prepare the ERA5 reanalysis data before combination with ISS-LIS lightning data. We extract the global 1-hourly
averaged value of the vertical velocity at the 450 hPa level between March 2017 and March 2018. We re-grid the data onto
a $2.50° \times 2.50°$ latitude and longitude grid because it is similar to that typically used in global chemistry climate models.
Second, we extract the value of the vertical velocity within this grid for each of the lightning flashes reported by ISS-LIS.
Third, we create groups of all the lightning flashes that coincide in each grid cell, i. e., we define groups of lightning flashes
taking place at the same hour and within a $2.50° \times 2.50°$ grid box. There is a unique value of the vertical velocity at 450 hPa
for each lightning flash in each group. Finally, we calculate the ratios of LCC(>9 ms)- and LCC(>18 ms)-lightning to typical
lightning flashes within each group of flashes.





We restrict our analysis to groups of flashes that include LCC(>9 ms)- and/or LCC(>18 ms)-lightning flashes and where the sign of the vertical velocity indicates upward transport of air. We assume that the non-observation of LCC-lightning flashes during the fast passage of ISS-LIS over the thunderstorm does not provide enough information to assume that the observed thunderstorm cannot produce LCC-lightning at all. Therefore, we do not include thunderstorms exclusively producing typical lightning flashes during the passage of ISS-LIS. We consider that grid cells where the movement of air is dominated by

downward velocity are not representative of thunderstorms. Applying these criteria, we find $1.6342 \times 10^4$ and $2.981 \times 10^3$ groups of flashes including LCC(>9 ms)-lightning and LCC(>18 ms)-lightning, respectively. We plot in Fig. 2 the obtained ratios of LCC(>9 ms)- and LCC(>18 ms)-lightning to typical lightning flashes versus the updraught mass flux, estimated as the vertical velocity divided by the acceleration of gravity ($9.8$ m s$^{-2}$). The high dispersion of values shown in Fig. 2 indicates that a possible relationship between the ratios of LCC(>9 ms)- and LCC(>18 ms)-lightning to typical lightning flashes is not

obvious.

Next, we analyze the data presented in Fig. 2. The average value of the updraught mass flux for the studied thunderstorms is $0.108$ kg m$^{-2}$ s$^{-1}$. Most of the studied thunderstorms have updraught mass fluxes below $0.2$ kg m$^{-2}$ s$^{-1}$. In particular, only 6.9% of the thunderstorms included in the left panel have updraught mass fluxes larger than $0.2$ kg m$^{-2}$ s$^{-1}$, while this quantity is reduced to 0.5% for updraught mass fluxes larger than $0.5$ kg m$^{-2}$ s$^{-1}$. In the right panel, only 5% and 1.5%

of the included thunderstorms have updraught mass fluxes larger than 0.2 and 0.3 kg m$^{-2}$ s$^{-1}$, respectively. In an effort to develop a parameterization of the ratio of LCC-lightning to typical lightning that is not over-represented by points of Fig. 2 with updraught mass fluxes below $0.2$ kg m$^{-2}$ s$^{-1}$, and which is also applicable for projected simulations, we apply a discrete binning of the data using a $2.5 \times 10^{-3}$ kg m$^{-2}$ s$^{-1}$ window. Red lines of Fig. 3 show the corresponding binned data.

The binned data shown in Fig. 3 (red lines) indicate a possible quadratic relationship between the updraught mass flux and the

ratios of LCC(>9 ms)- and LCC(>18 ms)-lightning to typical lightning flashes below ~0.5 and ~0.3 kg m$^{-2}$ s$^{-1}$, respectively. Due to the lack of points above these values of the updraught mass flux, the binned data are noisy. Therefore, we approximate the ratio of LCC(>9 ms)-lightning to typical lightning with a quadratic model between 0 and $0.5$ kg m$^{-2}$ s$^{-1}$ (blue line in the first panel of Fig. 3). We obtain that the ratio of LCC(>9 ms)-lightning to typical lightning ($R_9$) can be calculated as

$$R_9 = -5.12 \times M^2 + 2.77 \times M + 0.05, \tag{1}$$

where $M$ correspond to the updraught mass flux.

In the same manner, we approximate the ratio of LCC(>18 ms)-lightning with typical lightning to a quadratic model between 0 and $0.3$ kg m$^{-2}$ s$^{-1}$ (blue line in the second panel of Fig. 3). We obtain that the ratio of LCC(>18 ms)-lightning to typical lightning ($R_{18}$) can be calculated as

$$R_{18} = -8.01 \times M^2 + 3.02 \times M - 0.004. \tag{2}$$

Binning the data brings a degree of arbitrariness into the model. Therefore, we compare the obtained quadratic approximation of the binned data to a cubic smoothing spline fitting over the original data using the function `UnivariateSpline` of





`Scipy` (Virtanen et al., 2020) and setting a weight equal to the inverse of the standard deviation of the data and a smoothing factor equal to the number of observations, satisfying the Boor's criterion (De Boor and De Boor, 1978). The obtained cubic smoothing splines are shown as green lines in Fig. 3. Comparison of the quadratic and the cubic smoothing spline models in

both panels of Fig. 3 shows a good agreement between both models below 0.5 kg m$^{-2}$ s$^{-1}$. Therefore, we use the quadratic model to implement the ratios of LCC(>9 ms)- and LCC(>18 ms)-lightning to typical lightning flashes in LNOX.

### 3.3 Implementation in the MESSy submodel LNOX

In this section we describe the implementation of the LCC-lightning parameterizations described by equations (1) and (2) as a new subroutine called `lcc` in the `LNOX` submodel.

The new `lcc` subroutine receives the updraught mass flux (in kg m$^{-2}$ s$^{-1}$) and the lightning flash frequency (in s$^{-1}$) as inputs. Using equations (1) and (2) and a scaling factor that depends on the time step and the spatial resolution, the `lcc` subroutine calculates the ratios of LCC(>9 ms)- and LCC(>18 ms)-lightning to typical lightning flashes (set to zero if they are negative). The scaling factor is defined as a control namelist parameter of LNOX. The subroutine calculates the LCC(>9 ms)- and LCC(>18 ms)-lightning flash frequencies by multiplying the lightning flash frequency by the calculated ratios. The outputs

of the `lcc` subroutine are the LCC(>9 ms)- and LCC(>18 ms)-lightning flash frequencies and densities in units s$^{-1}$ and s$^{-1}$m$^{-2}$, respectively.

Finally, we define four new channel objects (see Jöckel et al. (2010)) in the `LNOX` submodel, namely the LCC(>9 ms)- and LCC(>18 ms)-lightning flash frequencies (s$^{-1}$) and densities (in s$^{-1}$ m$^{-2}$), in order to extract the LCC(>9 ms)- and LCC(>18 ms)-lightning flash occurrence from the simulations.

## 4 Example application

One year simulation was carried out for a demonstration of the developed LCC(>9 ms)- and LCC(>18 ms)-lightning parameterizations. The simulation setup is described in section 4.1. The obtained lightning flash frequency resulting from each parameterization is presented in section 4.2. Finally, the LCC-lightning flash frequency is presented in section 4.3, including a comparison with observational data from ISS-LIS.

### 4.1 Simulation setup

In this example, we apply EMAC in the T42L90MA resolution, i.e. with a quadratic Gaussian grid of 2.8° × 2.8° in latitude and longitude with 90 vertical levels reaching up to the 0.01 hPa pressure level and with 720 s time step length (Jöckel et al., 2016). We employ the namelist setup for purely dynamical simulations (referred to the E5 setup, no chemistry) in the mode of free running simulation. We use the Tiedtke convection scheme (Tiedtke, 1989) implemented in the submodel CONVECT.

The simulation period is the same as that used to develop the LCC-lightning parameterization, i.e., between 1 March, 2017 and 28 February, 2018. However, we start the simulation on January, 2017 using ERA-Interim reanalysis meteorological fields (ECMWF, 2011) as initial conditions and considering three months of spin-up time. The lightning flash density, LCC-lightning





**Table 1.** Lightning scaling factor used for each lightning parameterization.

| Lightning parameterization | Reference | Proxy | Scaling factor |
|---|---|---|---|
| $P_{cth}$ | Price and Rind (1992) | Cloud top height | 6.798 |
| $L_{cth}$ | Luhar et al. (2021) | Cloud top height | 3.882 |
| $G_{updr}$ | Grewe et al. (2001) | Updraft velocity | 3.815 |
| $A_{prec}$ | Allen and Pickering (2002) | Convective precipitation | 0.057 |
| $A_{updr}$ | Allen and Pickering (2002) | Updraft strength at 440 hPa | 0.093 |
| $P_{cth} + A_{prec}$ | Price and Rind (1992); Allen and Pickering (2002) | Cloud top height and convective precipitation | 1.130 |

flash frequencies and LCC-lightning flash densities are output every 5 hour. We do not modify the lightning-produced $NO_x$ in the code, as to the best of our knowledge there are no investigations reporting a difference in the production of $NO_x$ by

LCC-lightning with respect to typical lightning.

We need to find the upward mass flux scaling factor we have to use to implement equations (1) and (2) in our in T42L90MA resolution simulations. The upward mass flux averaged over a grid cell is influenced by the total area of the cell, the time step and the vertical resolution. In fact, as reported by Tost et al. (2007), lightning parameterizations based on the vertical velocity have to be re-scaled for different vertical resolutions. Therefore, we compare the maximum instantaneous value of the upward

mass flux extracted from one year EMAC simulation with the maximum value of the upward mass flux in a $2.5° \times 2.5°$ latitude and longitude box from ERA5. We find that the maximum upward mass flux extracted from ERA5 is 6.57 times higher than the upward mass flux extracted from EMAC using different horizontal and vertical resolutions. Therefore, we use 6.57 as a upward mass flux scaling factor to multiply the upward mass flux before using equations (1) and (2) for the T42L90MA resolution. For the T52L41DLR resolution (41 vertical levels), we obtain an upward mass flux scaling factor of 6.95.

**4.2 Lightning flash frequency**

As explained in section 3.3, the developed LCC-lightning parameterizations are based on the lightning parameterization included in the atmospheric model. Therefore, we analyze the lightning density obtained with each of the employed lightning parameterizations first.

We have used a lightning scaling factor for each lightning parameterization in order to fix the annual global lightning flash

rate to 45 flashes per second (Christian et al., 2003; Cecil et al., 2014). The lightning scaling factors are shown in Table 1.

The upper panel of figure 4 shows the global annual average flash density provided by OTD/LIS from 4 May 1995 to 31 December 2014 (Christian et al., 2003; Cecil et al., 2014), while the rest of panels of figure 4 show the simulated global annual average flash density using different lightning parameterizations. According to space-based observations, the land/ocean contrast is nearly $3 : 1$ (Christian et al., 2003; Cecil et al., 2014; Blakeslee et al., 2020). As previously reported by Tost et al.

(2007), $P_{cth}$ underestimates the lightning flash density over the oceans, producing a land/ocean contrast of about $5 : 1$. $G_{updr}$, $P_{cth} + A_{prec}$ and $L_{cth}$ overestimate the lightning flash density over the ocean, producing a contrast of about $1 : 1$. Finally, we obtain the highest overestimation over the ocean using $A_{prec}$ and $A_{updr}$, obtaining contrasts of $2 : 3$ and $4 : 1$, respectively.





**Table 2.** Contrast between land and ocean for LCC(>9 ms)-, LCC(>18 ms)- and typical-lightning flashes for each lightning parameterization.

| Lightning parameterization | Lightning land/ocean contrast | LCC(>9 ms)-lightning land/ocean contrast | LCC(>18 ms)-lightning land/ocean contrast | Ratio LCC(>9 ms)/typical lightning | Ratio LCC(>18 ms)/typical lightning |
|---|---|---|---|---|---|
| Observed | $3:1$ | $2:1$ | $2:1$ | $66 \times 10^{-3}$ | $8 \times 10^{-3}$ |
| $P_{cth}$ | $5:1$ | $5:1$ | $10:3$ | $63 \times 10^{-3}$ | $8 \times 10^{-3}$ |
| $L_{cth}$ | $1:1$ | $1:1$ | $2:3$ | $65 \times 10^{-3}$ | $9 \times 10^{-3}$ |
| $G_{updr}$ | $1:1$ | $7:3$ | $1:4$ | $49 \times 10^{-3}$ | $1 \times 10^{-3}$ |
| $A_{prec}$ | $2:3$ | $2:3$ | $1:4$ | $58 \times 10^{-3}$ | $4 \times 10^{-3}$ |
| $A_{updr}$ | $4:1$ | $2:3$ | $1:9$ | $54 \times 10^{-3}$ | $2 \times 10^{-3}$ |
| $P_{cth} + A_{prec}$ | $1:1$ | $1:1$ | $1:1$ | $62 \times 10^{-3}$ | $6 \times 10^{-3}$ |

Figure 4 shows that the lightning parameterization can significantly influence the simulated spatial and seasonal distribution of lightning flashes. Therefore, we expect that the choice lightning parameterization affects the simulated LCC-lightning flash climatology.

### 4.3 LCC-lightning flash frequency

Figures 5 and 6 show the simulated annual average LCC(>9 ms)- and LCC(>18 ms)-lightning flash density using different lightning parameterizations, while the columns 2, 3 and 4 of Table 2 indicate the contrast between land and ocean for LCC(>9 ms)-, LCC(>18 ms)- and typical-lightning flashes. In the case of the $P_{cth}$ and the $L_{cth}$ lightning parameterizations, the spatial distributions of LCC(>9 ms)-lightning flash densities are nearly similar to the corresponding spatial distributions of lightning flash density. However, the land/ocean contrast is slightly shifted towards ocean in the case of LCC(>18 ms)-lightning flash density with respect to lightning density (from $5:1$ to $10:3$). In the case of $G_{updr}$, the contrast land/ocean is significantly shifted towards land for LCC(>9 ms)-lightning (from $1:1$ to $7:3$) and shifted towards ocean for LCC(>18 ms)-lightning (from $1:1$ to $2:8$) with respect to lightning density. The global distribution of LCC(>18 ms)-lightning density is substantially different to the spatial distributions of typical and LCC(>9 ms)-lightning densities, with maximum LCC(>18 ms)-lightning activity at higher latitudes (North and South) and in Southern Asia. The contrast land/ocean is $1:1$ for LCC(>9 ms)- , LCC(>18 ms)- and typical-lightning flash densities when using the $P_{cth}+A_{prec}$ lightning parameterization. The $A_{prec}$ parameterization produces a land/ocean contrast shifted towards ocean for LCC(>9 ms)- and LCC(>18 ms)-lightning densities with respect to the land/ocean contrast of lightning flash density. In the case of $A_{updr}$, we obtain a significant shift to ocean in the land/ocean contrast for LCC(>9 ms)- and LCC(>18 ms)-lightning with respect to typical lightning (from $4:1$ to $2:3$ and to $1:9$, respectively).

Next, we compare the simulated and the observed ratios of LCC(>9 ms)- and LCC(>18 ms)-lightning to typical lightning. As detailed in section 4.2, the ratio of LCC(>9 ms)-lightning to all lightning flashes at a global scale reported by ISS-LIS is about $6.6 \times 10^{-2}$, while the ratio of LCC(>18 ms)-lightning to all lightning flashes at a global scale is about $8 \times 10^{-3}$. The last two columns of Table 2 show the simulated globally averaged ratios of LCC(>9 ms)- and LCC(>18 ms)-lightning to typical lightning using different lightning parameterizations. The best agreement between the observed and the simulated ratios are obtained within lightning parameterizations based on the CTH, such as $P_{cth}$, $L_{cth}$ and $P_{cth}+A_{prec}$.





The seasonal observed and simulated ratios of LCC(>9 ms)-lightning and LCC(>18 ms)-lightning to typical lightning are shown in Figure 7 and 8, respectively. Lightning data has been gridded in $2.8° \times 2.8°$ in latitude and longitude, while differences between the seasonal observed and simulated ratios of LCC(>9 ms)-lightning to typical lightning are shown in Figures 9-

12. We include in Figures 9-12 the globally averaged difference and the spatial correlation coefficients between observation and simulations ($r$). In general, all the investigated lightning parameterizations produce a fairly good estimation of the ratio of LCC(>9 ms)-lightning to typical lightning in Central Africa, where the observed ratio reaches its minimum and non-negligible value. However, they tend to underestimate the ratio over the oceans, where the observed ratio reaches its maximum values. Finally, all the parameterizations tend to overestimate the ratio over South America, especially over the Eastern coast. Disagree-

ment between the observed and the modeled ratio of LCC(>9 ms)-lightning to typical lightning in the Eastern coast of South America can be due to the South Atlantic Anomaly (SAA). As reported by Buechler et al. (2014), high noise rates are frequent in LIS observations over the SAA. The noise can cause LIS missing the tail of the optical signal emitted by LCC-lightning.

Due to the lack of observations, comparison between simulated and observed spatial distributions of the ratio of LCC(>18 ms)-lightning to typical lightning is not so straightforward as in the case of LCC(>9 ms)-lightning. However, Figure 8 indicates

that simulated and observed spatial distribution of the ratio of LCC(>18 ms)-lightning to typical lightning are nearly in agreement. The simulation tends to underestimate the ratio of f LCC(>18 ms)-lightning to typical lightning over South America, the eastern coast of North America, Central Africa and Southeastern Asia.

The upper panel of Figure 13 shows the seasonal evolution of the spatial correlation coefficient between observations and simulations for land and ocean. The correlation coefficient over land ranges between 0.2 and 0.5, without showing significant

differences between each of the used lightning parameterizations. The correlation coefficient over ocean oscillates between 0.1 and 0.2. The lightning parameterization that produces the higher correlation coefficient is $P_{cth}+A_{prec}$, while the one producing the lower correlation coefficient is $A_{updr}$. The highest correlation over land is reached during June, July and August, in coincidence with the maximum lightning activity over the Northern Hemisphere. On the contrary, the higher correlation coefficient over ocean is reached during March, April and May.

Finally, the lower panel of Figure 13 shows the seasonal global correlation coefficient versus the deviation averaged over all the grid cells between observations and simulations for the ratio of LCC(>9 ms)-lightning to typical lightning. The best globally averaged agreement between observations and simulations is produced by the $P_{cth}+A_{prec}$ and $A_{prec}$ lightning parameterizations, while the worst globally averaged agreement are produced by the $A_{updr}$ lightning parameterization. The $L_{cth}$ lightning parameterization produces a better agreement with observations than $P_{cth}$. Finally, both $L_{cth}$ and $P_{cth}$ produces a

better estimate of the globally averaged ratio of LCC(>9 ms)-lightning to typical lightning than $G_{updr}$.

## 5   Discussion

In this section, we analyze the seasonal and spatial distribution of the ratio of LCC(>9 ms)- and LCC(>18 ms)-lightning to typical lightning by comparing with observation.





**Table 3.** Indication of the observed ratio of LCC(>9 ms)/typical lightning by region and season. High, medium and low values correspond to approximately values greater than $10^{-1}$, between $3 \times 10^{-2}$ and $10^{-1}$, and lower than $3 \times 10^{-2}$, respectively. The symbol - represents no data.

| Region | DJF | MAM | JJA | SON |
|---|---|---|---|---|
| Northern America | **High** | **High** | **High** | **High** |
| South America | Medium | Medium | Medium | Medium |
| Caribbean | Medium | Medium | Medium | **High** |
| Central America | Medium | Medium | Medium | Medium |
| Middle Africa | Low | Low | Low | Medium |
| Eastern Africa | Low | Low | - | Medium |
| Western Africa | Medium | Low | Low | Low |
| Southern Africa | **High** | **High** | **High** | **High** |
| Northern Africa | - | - | Low | - |
| Southern Europe | Medium | **High** | Medium | **High** |
| Western Europe | Low | Medium | Medium | Medium |
| Eastern Europe | - | Medium | Medium | - |
| Western Asia | - | Low | Low | Medium |
| Central Asia | - | Medium | Medium | - |
| Southern Asia | Medium | Medium | Medium | Medium |
| Eastern Asia | Medium | Medium | Medium | Medium |
| Southeastern Asia | **High** | Medium | | **High** |
| Micronesia | Medium | Medium | Medium | Medium |
| Melanesia | **High** | **High** | **High** | **High** |
| Australia and New Zealand | **High** | **High** | Medium | **High** |
| Atlantic Ocean | Low | **High** | **High** | **High** |
| Indian Ocean | Low | **High** | Medium | **High** |
| Pacific Ocean | Medium | **High** | **High** | **High** |

## 5.1 Seasonality in the ratio of LCC(>9 ms)-lightning to typical lightning

In this section we put the results into content by analyzing the seasonal and regional distribution of the ratio of LCC(>9 ms)-lightning to typical lightning showed in Figure 7. We indicate in Table 3 the relative value of the observed ratio of LCC(>9 ms)/typical lightning by region and season. The ratio of LCC(>9 ms)-lightning to typical lightning is high in regions downwind of the continents, which are known to be the preferred regions where cyclones evolve (Eckhardt et al., 2004). In the so-called warm conveyor belt of the cyclones, a broad band of air masses are rapidly ascending from lower levels to higher levels causing
instability and the development of deep convection. Deep convection developing in warm conveyor belts is generally weaker than in pre-frontal convective systems, supporting the development of LCC-lightning. The LCC-lightning parameterization developed here reproduces well the observed ratio of LCC(>9 ms)/typical lightning in regions for intercontinental transport of trace gases with a high occurrence of warm conveyor belts.





However, the oceanic region influenced by the outflow from West Africa to South America is not commonly influenced

by warm conveyor belts. Intercontinental transport of aerosols and trace gases is commonly observed in this region (Ansmann et al., 2009). The parameterization of LCC-lightning developed in this study significantly underestimates the ratio in the outflow from West Africa (see Figure 9). This indicates that the aerosols in regions for intercontinental transport of trace gases can play a role in the occurrence of LCC-lightning. This is not surprising, as it is known that aerosols participate in the electrification of thunderstorms (Tao et al. (2012); Pérez-Invernón et al. (2021a) and references therein).

The simulations suggests seasonality in the ratio of LCC(>9 ms)-lightning to typical lightning over the large oceans (Atlantic, Indian and Pacific Oceans). Figure 7 shows that the highest oceanic ratios of LCC(>9 ms)-lightning to typical lightning are reached in winter thunderstorms, characterized by weak updrafts, while intermediate values of the ratio are reached during MAM and SON seasons. This seasonality is partially in agreement with observations. The observed spatial distributions of the ratio of LCC(>9 ms)-lightning to typical lightning over the oceans are in agreement with simulations in MAM and SON, when

the updraft reaches intermediate values. However, there is not a good agreement between the simulations and observations over the ocean in DJF and JJA, when the updraft have more extreme (low and high) values.

During the season including December, January and February (DJF) lightning activity is mostly focused on the Southern Hemisphere (Blakeslee et al., 2020), the Gulf of Mexico, the Tornado Alley of the United States, the Mediterranean Sea and Northern India. Figure 9 shows that all the lightning parameterization underestimate the ratio of LCC(>9 ms)-lightning to

typical lightning in the Tornado Alley of the United States, the Eastern coast of Australia, the Eastern coast of South Africa and the South Pacific Ocean, while they overestimate the ratio in the Mediterranean Sea. In addition, $P_{cth}$, $L_{cth}$ and $P_{cth}+A_{updr}$ slightly overestimate the ratio in the central part of South America and in Northern India.

Lightning activity during March, April and May (MAM) is distributed all over the globe (Blakeslee et al., 2020). Apart from the active regions during the winter season in the North Hemisphere, the spring season presents high lightning activity over

India, Eastern and Western Asia, central and Eastern North America, central Europe and the Northwestern Atlantic Ocean. Figure 10 shows that all the lightning parameterizations underestimate the ratio of LCC(>9 ms)-lightning to typical lightning mainly over the Northwestern Atlantic Ocean, the North Indian Ocean, the Eastern coast of South Africa, the Melanesia and the Eastern coast of Asia. On the contrary, they tend to overestimate the ratio mainly in the Southwestern Atlantic Ocean. $P_{cth}$, $L_{cth}$ and $P_{cth}+A_{updr}$ also overestimate the ratio in Northwestern America, central Europe and central Asia.

During June, July and August (JJA) lightning activity is focused in the Northern Hemisphere (Blakeslee et al., 2020), especially over North and Central America, Europe, Central and North Africa and the main part of Asia. There are also some regions with high lightning activity at the Eastern coast of South America, in the North of South America and at the Eastern coast of South Africa. Figure 11 shows that all the lightning parameterizations underestimate the ratio over the Northwestern Atlantic Ocean, the Pacific Ocean, the North of Canada and the coast of South Africa. All the lightning parameterizations

overestimate the ratio over central and Eastern Asia and in the Eastern coast of South America.

The global distribution of lightning activity during September, October and November (SON) is nearly similar as during MAM (Blakeslee et al., 2020). Figure 12 shows that the tested lightning parameterizations underestimate the ratio of





LCC(>9 ms)-lightning to typical lightning mainly over the the Northwestern Atlantic Ocean, the Mediterranean Sea, Australia and Southeastern Asia. $P_{cth}$, $L_{cth}$ and $P_{cth}+A_{updr}$ overestimate the ratio mainly in the Eastern coast of South America.

## 5.2 Seasonality in the ratio of LCC(>18 ms)-lightning to typical lightning

All lightning parameterizations overestimate LCC(>18 ms)-lightning over ocean. However, the low lightning activity over ocean in conjunction with the low global ratio of LCC(>18 ms)-lightning to typical lightning entail very few observations of LCC(>18 ms)-lightning over ocean, making it difficult to compare the simulated and the observed spatial distributions over ocean. Therefore, we focus the discussion on seasonal results only on the spatial distribution over land and only using $P_{cth}$, 345 $L_{cth}$, $A_{prec}$ and $P_{cth}+A_{prec}$.

During December, January and February a good agreement between the simulated and the observed ratio of LCC(>18 ms)-lightning to typical lightning is achieved in South America, Central and South Africa, Southeastern Asia, Australia, the Gulf of Mexico and the Tornado Alley. These are the regions with the highest lightning activity during DJF. However, the simulated ratio is higher than the observed in Western Europe.

In the months March, April and May there is a good agreement between simulations and observations of the ratio of LCC(>18 ms)-lightning to typical lightning in South America, Central and South Africa, Central and Eastern North America, Europe, Southeastern Asia, Eastern and Southern Asia. The simulated ratios are higher than observed in South Africa, Northwestern America, Australia, Central Asia and Eastern Europe, while they are lower in Central Africa.

During June, July and August there is a good agreement between simulations and observations of the ratio of LCC(>18 ms)-355 lightning to typical lightning in the Northern Hemisphere. The ratio is overestimated in South Australia.

During September, October and November there is a good agreement between simulations and observations of the ratio of LCC(>18 ms)-lightning to typical lightning in Eastern North America, South Europe, central and Eastern South America, Africa, Australia and Southern Asia. The ratio is overestimated in Western North America, Central and Eastern Europe and the Southern South America.

Finally, we have performed a simulation of years 2009, 2010 and 2011 to compare the ratio of LCC(>18 ms)-lightning to typical lightning using the $P_{cth}+A_{prec}$ lightning parameterization. We have found global annual ratios of LCC(>18 ms)-lightning to typical lightning of $6.57 \times 10^{-3}$, $6.96 \times 10^{-3}$ and of $6.58 \times 10^{-3}$ for 2009, 2010 and 2011, respectively. Therefore, we conclude that there are not large differences in other years.

## 6 Conclusions

We have developed for the first time two parameterizations that use the updraft strength at 450 hPa pressure level as a proxy for the ratio of LCC(>9 ms)- and LCC(>18 ms)-lightning to typical lightning, respectively. We have implemented these parameterizations as an upgrade of the `LNOX` submodel of the Modular Earth Submodel System (from v2.54 onwards) and made it available for the community MESSy concept. We have run a one-year simulation with EMAC using different lightning parameterizations to calculate the total lightning. The obtained global ratio of LCC(>9 ms)- and LCC(>18 ms)-lightning to typical



total lightning using a set of 6 lightning parameterizations are in agreement with the ratio reported by ISS-LIS. However, the simulated spatial distribution of the ratios strongly depends on the choice of the lightning parameterization. We found that the best agreement between the observed and the simulated spatial distributions of the ratios on a seasonal basis is achieved when using a novel combined lightning parameterization based on the cloud top height ($P_{cth}$) over land and on the convective precipitation ($A_{prec}$) over ocean.

The lower lightning frequency over ocean than over land entail a significantly lower total amount of observations of LCC(>18 ms)-producing thunderstorms over ocean. Since LCC(>18 ms)-lightning flashes are rare, the climatology of LCC(>18 ms)-lightning provided by ISS-LIS over the oceans is imprecise and rough. Therefore, a correct comparison of the simulated and the observed climatology of LCC(>18 ms)-lightning is not realistic, while for LCC(>9 ms)-lightning we have enough data to receive realistic results.

Geostationary-based optical instruments devoted to monitor the occurrence of lightning, such as the Geostationary Lightning Mapper (GLM) aboard the Geostationary Operational Environmental Satellite-16 (GOES-16) since 2017 (Goodman et al., 2013), the Lightning Mapping Imager (LMI) aboard the Feng-Yun-4 satellite (FY-4) since 2018 (Yang et al., 2017), and the launch of the Meteosat Third Generation (MTG) geostationary satellites of the EUropean organization for the exploitation of METeorological SATellites (EUMETSAT) equipped with a Lightning Imager (LI) in 2022 (Stuhlmann et al., 2005) will

provide new observations that will serve to complement the global climatology of LCC-lightning provided by TRMM-LIS and ISS-LIS. New data from these instruments will serve to improve the parameterizations of LCC-lightning presented here.

Future work on investigating the relationships between lightning-ignited wildfires and LCC-lightning can serve to use LCC-lightning parameterizations (as the one presented here) as a proxy for lightning-ignited wildfires in forecasting or global atmospheric models. However, more reports about the optical and/or ELF signal emitted by fire-igniting lightning are needed

to confirm the role of LCC-lightning in the production of lightning-ignited fires.

In addition, the simultaneous observations of lightning and sprites by space-based instruments can serve to develop a new parameterization of sprites based on LCC-lightning parameterizations. A parameterization of sprites in global chemistry–climate models can be employed to investigate the role of sprites in the chemistry of the mesosphere . The Modular Multispectral Imaging Array (MMIA) onboard the Atmosphere–Space Interactions Monitor (ASIM) (Neubert et al., 2019; Chanrion et al.,

2019) since 2018 is equipped with three photometers that can simultaneously report the occurrence of sprites and the duration of the optical signal emitted by the lightning-parent, providing us new relationships between LCC-lightning and sprites.

*Data availability.* All data used in this paper are directly available after a request is made to authors F. J. P. I (FranciscoJavier.Perez-Invernon@dlr.de) or H. H. (Heidi.Huntrieser@dlr.de). The ERA5 meteorological data are freely accessible through Copernicus Climate Change Service (C3S) (2017): ERA5: Fifth generation of ECMWF atmospheric reanalyses of the global climate Copernicus Climate Change

Service Climate Data Store (CDS) (https://cds.climate.copernicus.eu/cdsapp). ISS-LIS data can be freely downloaded from https://ghrc.nsstc.nasa.gov/lightning/data/data_lis_iss.html, DOI: http://dx.doi.org/10.5067/LIS/ISSLIS/DATA107. The data of the simulations presented in this study are freely available under https://doi.org/10.5281/zenodo.5606230 (Pérez-Invernón, F. J. et al., 2021).



*Code availability.* The Modular Earth Submodel System (MESSy) is continuously developed and applied by a consortium of institutions. The usage of MESSy and access to the source code is licensed to all affiliates of institutions which are members of the MESSy Consortium.

Institutions can become a member of the MESSy Consortium by signing the MESSy Memorandum of Understanding. More information can be found on the MESSy Consortium website (http://www.messy-interface.org, last access: 05 10 2021). As the MESSy code is only available under license, the code cannot be made publicly available. The parameterization of LCC-lightning has been developed based on MESSy version 2.54 and is included in version 2.55.

*Author contributions.* F.J.P.I.: Conceptualization, methodology, validation, formal analysis, investigation, data curation, writing—original

draft. H.H., P. J. and F.J.G.V.: Conceptualization, methodology, validation, formal analysis, supervision, investigation, writing—review and editing.

*Competing interests.* Authors declare no competing interests.

*Acknowledgements.* The authors would like to thank NASA for providing ISS-LIS lightning data and ECMWF for providing the data of ERA5 forecasting models. The EMAC simulations have been performed at the German Climate Computing Centre (DKRZ) through support

from the Bundesministerium für Bildung und Forschung (BMBF). DKRZ and its scientific steering committee are gratefully acknowledged for providing the HPC and data archiving resources.Authors would also like to thank Roland Eichinger (Deutsches Zentrum für Luft-und Raumfahrt, DLR) for providing valuable comments on this manuscript.

FJPI acknowledges the sponsorship provided by the Federal Ministry for Education and Research of Germany through the Alexander von Humboldt Foundation. Additionally, this work was supported by the Spanish Ministry of Science and Innovation, under projects PID2019-

109269RB-C43 and FEDER program. FJGV acknowledge financial support from the State Agency for Research of the Spanish MCIU through the 'Center of Excellence Severo Ochoa' award for the Instituto de Astrofísica de Andalucía (SEV-2017-0709).





**Figure 1.** Annual lightning flash density (first panel), LCC(>9 ms)-lightning flash density (second panel), ratio between LCC(>9 ms)-lightning and lightning flash densities (third panel) and ratio between LCC(>18 ms)-lightning and lightning flash densities (fourth panel) extracted from ISS-LIS data between March 2017 and March 2018 binned into 1° × 1° grids.



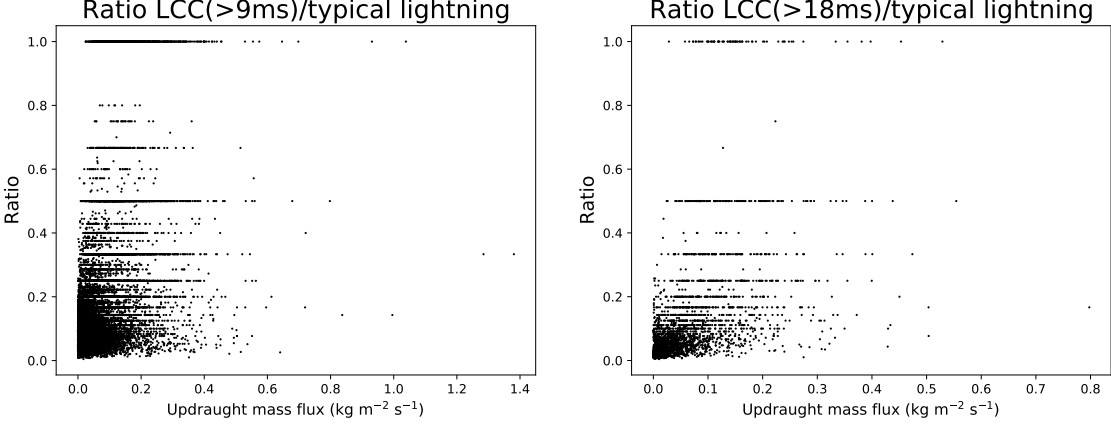

**Figure 2.** Ratios of (left panel) LCC(>9 ms)- and (right panel) LCC(>18 ms)-lightning to typical lightning flashes versus the updraught mass flux extracted from ERA5 1-hourly averaged $2.75° \times 2.75°$ grid cells for lightning reported by ISS-LIS globally between March 2017 and March 2018.





**Figure 3.** Ratios of (upper panel) LCC(>9 ms)- and (lower panel) LCC(>18 ms)-lightning to typical lightning flashes versus the updraught mass flux as in Fig. 2. We have added a cubic smoothing spline (green line), a quadratic fitting (blue line) and a binning (red line) to the data as described in the text.



**Figure 4.** (Upper panel) Global annual lightning observations by OTD/LIS using the OTD/LIS Gridded Lightning Climatology Data Collection, Version 2.3.2015, High Resolution Monthly Climatology (HRMC) from 4 May 1995 to 31 December 2014 (Christian et al., 2003; Cecil et al., 2014). As in Gordillo-Vázquez et al. (2019), the climatology has been degraded to 2.5°longitude × 1.9°latitude resolution. (Rest of panels) Simulated annual average flash density between March 1, 2017 and February 28, 2018 using different lightning parameterizations described in Table 1.





**Figure 5.** (Upper panel) Global annual lightning observations by OTD/LIS as in Figure 4. (Rest of panels) Simulated annual average LCC(>9 ms)-lightning flash density between March 1, 2017 and February 28, 2018 using different lightning parameterizations.



**Figure 6.** (Upper panel) Global annual lightning observations by OTD/LIS as in Figure 4. (Rest of panels) Simulated annual average LCC(>18 ms)-lightning flash density between March 1, 2017 and February 28, 2018 using different lightning parameterizations.





**Figure 7.** Seasonal observed (top panel) and simulated (rest of panels) ratio of LCC(>9 ms)-lightning to typical lightning using $P_{cth}$+$A_{prec}$ lightning parameterization. The maximum value of the colorbar is 0.44.

**Figure 8.** Seasonal observed (top panel) and simulated (rest of panels) ratio of LCC(>18 ms)-lightning to typical lightning using $P_{cth}$+$A_{prec}$ lightning parameterization. The maximum value of the colorbar is 0.33.







**Figure 9.** Difference between the observed and the simulated ratio of LCC(>9 ms)-lightning to typical lightning density December, January and February using different lightning parameterizations. The calculated global, land and ocean spatial correlation coefficients $r$ are included.



**Figure 10.** Difference between the observed and the simulated ratio of LCC(>9 ms)-lightning to typical lightning density March, April and May using different lightning parameterizations. The calculated global, land and ocean spatial correlation coefficients $r$ are included.







**Figure 11.** Difference between the observed and the simulated ratio of LCC(>9 ms)-lightning to typical lightning density June, July and August using different lightning parameterizations. The calculated global, land and ocean spatial correlation coefficients $r$ are included.



**Figure 12.** Difference between the observed and the simulated ratio of LCC(>9 ms)-lightning to typical lightning density September, October and November using different lightning parameterizations. The calculated global, land and ocean spatial correlation coefficients $r$ are included.



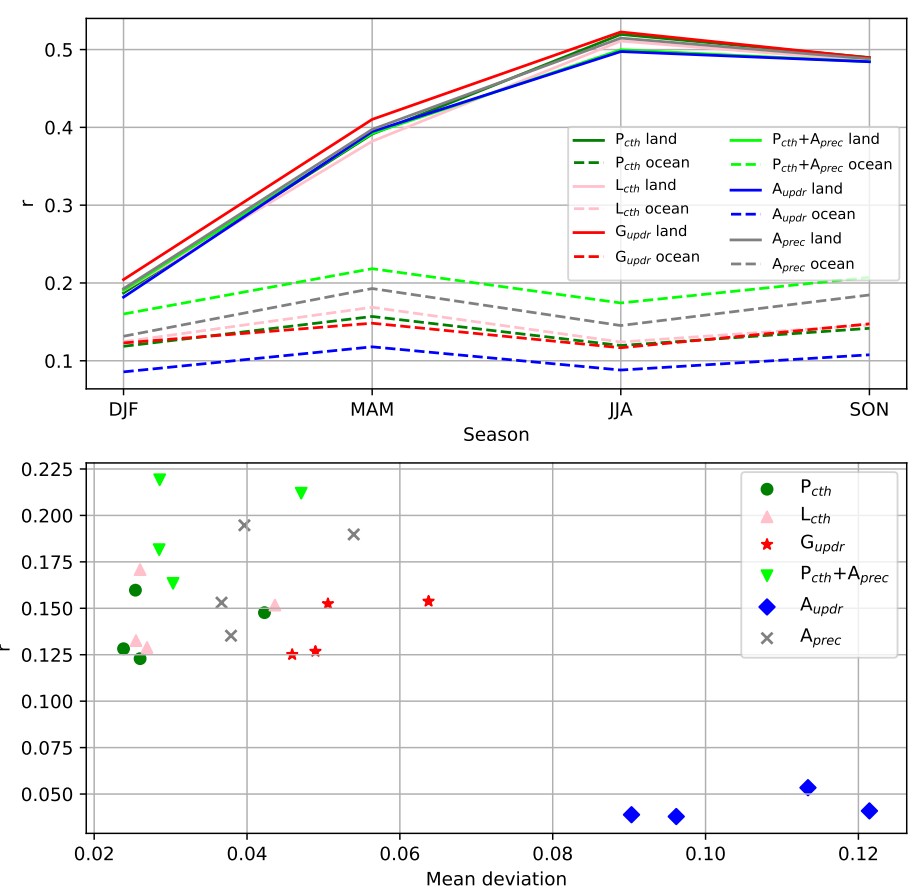

**Figure 13.** Seasonal evolution of the spatial correlation coefficient ($r$) between observed and simulated ratio of LCC(>9 ms)-lightning to typical lightning for land and ocean (upper panel) and seasonal global correlation coefficient versus the deviation averaged over all the grid cells between observed and simulated ratio of LCC(>9 ms)-lightning to typical lightning (lower panel). Each point represents a different season.





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
