# Peer review of "A parameterization of Long-Continuing-Current (LCC) lightning in the lightning submodel LNOX (version 3.0) of the Modular Earth Submodel System (MESSy, version 2.54)"

_Geoscientific Model Development, 2021_

## Referee Comment (RC2)

**General Comments**

Pérez-Invernón et al. utilized the Lightning Imaging Sensor to develop different Long-Continuing-Current (LCC) parameterizations and compared the simulations with observations. This new parameterization could benefit both LCC simulations and lightning NOx studies.

**Specific Comments**

1. L12: What is the meaning of typical lightning? It should be typical total lightning according to the main text. If I am wrong, please correct me.

2. L66-69: It is necessary to explain why do you only mention LCC (>9 ms) and LCC (>18 ms). How about the longer LCC?

3. L77: If I understand correctly, it is the total lightning distribution which agrees with that derived in Blakeslee et al. (2020). Please rephrase the sentence to make it clear and connect with the peak flash density and land-ocean contrast mentioned later.

4. L81-92: It would be interesting to see more discussion about the meaning of the ratios between LCC and total lightning flashes. While the introduction section has explained why the ocean has a larger ratio, how about the cause and meaning of the maximums over land? Are these also weak convection?

5. L94-95: It seems the ratio of LCC (>18 ms) to total lightning usually exists over the cells with large ratio of LCC (> 9ms) to total lightning. This indicates that LCC (>18 ms) is the subset of LCC (>9 ms) and explains the smaller number of LCC (>18 ms).

6. L113: As discussed in [Romps, D. M. (2019)] (https://doi.org/10.1029/2019GL085748), "the mixed-phase cloud region is bounded by the 273-K isotherm (where ice can first form) and the 240-K isotherm (where liquid drops freeze homogeneously)". They use IFluxT, defined to be the convective ice flux on the 260-K isotherm, which lies within the mixed-phase regions of clouds and is close to the 440-mbar isobar in a modern-day tropical sounding. I know it is a large work to do the sensitivity test, could authors point the importance of isotherm?

7. L127-L128: Shouldn't the tropospheric and middle atmosphere processes include the effects of anthropogenic emissions? maybe authors want to emphasize the meteorological atmosphere processes?

8. L143: How did the authors re-grid the updraft? It is better to use the maximum updraft in the grid according to the updraft references.

9. L159: Does "the possible relationship" stands for the relationship between ratio and updraught mass flux?

10. L171: It seems there are few points when the updraught mass flux larger than 0.3. Could this affect the regression?

11. L185: Did the authors get some model grids with updraught mass flux larger than 0.5? If so, the uncertainty of LNOx will also be large. If the authors have written some other manuscript implementing this parameterization, please tell readers the limitation.

12. L235-L236: As mentioned in Luhar et al. (2021), their marine parameterisation yields flash rates that are approximately an order of magnitude smaller than the PR92. Why the authors get the larger difference (5:1 and 1:1)? Is it caused by the model resolution?

13. L268: Are "maximum values" compared with both land and ocean data?

14. L282-L284: Interesting results of seasonal correlation. Do authors have any explanations?

15. L315-L316: Are extreme values of updraft from reanalysis or simulation? If they only exists in one dataset, that may explain the bad agreement.

16. In the Discussion section, authors usually use "good agreement", "higher", "lower" to explain the figures. They need to come up with a better way like putting some figures in the Supplements and add use some indexes to judge "good or bad" and "higher or lower".

**Technical Corrections**

1. L11: to find a global parameterization → to develop a global parameterization

2. L13: It is better to give the full name of EMAC in the Abstract

3. L31, L37: LSS are → LLS is

4. L42: Add a space between degree symbol and N(S)

5. L47: the process of separation of electrical charges → the process of electric charge separation

6. L51: could be helpful to ..... and (to) ....

7. L63: lightning with ... and (with) ...

8. L71: The TRMM-LIS ended in 2015 and a similar instrument onboard the ISS replaced it for a
   4 years mission ...

9. L76, L81, L93: It would be better to use a,b,c instead of first/second panel to point out the subplot. Please also check other figures.

10. L82: are also regions with → coincide with

11. L84: is clearer shown ... showing the ratio of → is more clearly shown by the ratio of .... in Fig 1c.

12. L89: between 35∘ N and 35∘ S latitude → between 35∘ N and 35∘ S

13. L90: All these regions are well-known regions for → All these regions are well-known for

14. L120: "section" → "Section"

15. L129: "by using" → "by"

16. L134: "a novel combination .... suggested by us" → "our novel combination"

17. L138: What is the definition of the "scaling factor" in Table 1? Because I do not see the same numbers in these references, authors may have their own definition.

18. L142: "prepare" → "process"

19. L143: The sentence can be simplified into "the global 1-hourly averaged values of the vertical velocity at the 450 hPa level between March 2017 and March 2018 are re-gridded onto a 2.50∘ × 2.50∘ latitude and longitude grid, which it is similar to that typically used in global chemistry climate models."

20. L197-L199: This is duplicated with L195-L196. If I misunderstand, please correct me.

21. L206: "with a quadratic Gaussian grid of 2.8∘ × 2.8∘ in latitude and longitude" → "with a .8∘ × 2.8∘ quadratic Gaussian grid"

22. L216: Please rephrase the sentence. The grammar is wrong.

23. L231-L232: "figure 4" → "Fig. 4"

24. L263: "Figure 7 and 8" → "Fig 7. and 8."

25. L264-L265: "Figures 9-12" → "Fig. 9-12"

26. L276: "off" → "of"

27. L300-L301: Please add the references.

28. L305: Which ratio?

29. L310: "suggests" → "suggest"

30. L317: "season" → "seasons"

31. L319: "parameterization" → "parameterizations"

32. L342: "entail" → "entails"

33. L375: "entail" → "entails"

34. L386: "that will serve to complement" → "that can complement"

35. L386: "will serve to improve the" → "will improve"

36. L387: "will serve to improve" → "is needed to improve"

37. L391: Too many "serve to" are used in the Conclusion. Please polish it.

38. Fig 13. "Each point represents a different season." It is better to use different opacity (or other symbols) for each point. Otherwise, readers do not know which point is which season.

---

## Author Comment (AC1)

We thank the reviewers for the time spent in the revision of this manuscript. In this document, we provide a detailed answer to all the comments raised by the reviewers (blue).

**Reviewer 1**

**Overview:**

Pérez-Invernón present a novel method of parameterizing the distribution of Long-Continuing Current (LCC) flashes within the framework of a chemistry and climate model. These flashes may have a disproportionate role in the ignition of wildfires and the triggering of sprites.

**General Comments:**

In general, the paper is well written although the number of figures could be reduced and the discussion section is a bit tedious.

We thank the reviewer for these encouraging comments and for the time spent in the revision of this manuscript.

Please note that we have significantly reduced the total number of figures and the discussion section following the reviewer comments.

**Specific Comments:**

L68-69: What is the rationale for using two thresholds for LCC flashes? Are LCC18 flashes much more likely to result in fires or sprites? Have others made this division? If no, perhaps only show one or two plots of them.

Let us first discuss the first threshold (LCC(>9 ms)-lightning). Bell et al. (1998) reported sprites triggered by CG lightning with intense continuing currents lasting ~1 ms. However, as explained by Bitzer (2017), LIS can detect individual discharges from an intracloud leader in as many as four consecutive frames [Brunner, 2016]. For this reason, Bitzer (2017) and our study use a minimum of of five consecutive frames to characterize continuing current. As explained by Bitzer (2017), "this methodology will not identify LCC-lightning with a short continuing current". However, "this minimizes the possibility that the data set contains noncontinuing current events".

Next, we discuss the second threshold (LCC(>18 ms)-lightning). Long-Continuing-Current Lightning (LCC-lightning) with duration between 40 ms and 282 ms have been proposed to be the main precursors of LIW Mceachron et al. (1942) and Fuquay et al. (1967). As reported by Bitzer (2017), the duration of the continuing phase detected by LIS should be considered a minimum. As an example, Bitzer (2017) compared the duration of the optical signal of a flash (7-9 ms) with the duration of the continuing current reported by the Huntsville Alabama Marx Meter Array (HAMMA) of 22 ms. Therefore, we consider that a flash has the potential to ignite fires if its optical emission is detected in twenty or more consecutive frames (LCC(>18 ms)-lightning flashes). According to the comparison of the continuing phase duration between the optical signal (7-9 ms) and the radio signal measured by HAMMA (22 ms) and reported by Bitzer (2017), LCC(>18 ms)-lightning flashes could have a continuing current lasting about 44-57 ms. This is consistent with the minimum duration of 40 ms for flashes that ignited fires reported by Mceachron et al. (1942) and Fuquay et al. (1967).

We have added this reasoning to the manuscript.

L111: Expand on why updrafts are especially important in determining the number of LCC flashes as opposed to the total number of flashes.

We have expanding the discussion on why we use the updrafts.

L115-117: Justify the use of a gridded 0.25 degree resolution vertical velocity product to parameterize flashes which occur at a higher spatial resolution. Is the product a one-hour average vertical velocity or an instantaneous value.

In this section we describe the meteorological data we extract from ERA5 reanalysis. In Section 3.2 we explain how we use the data to develop the parameterization of LCC-lightning. As described in Section 3.2, we re-grid the data onto a 2.5 x 2.5 degree grid.

The vertical velocity of the ERA5 reanalysis product is an instantaneous parameter (see Table 9 in https://confluence.ecmwf.int/display/CKB/ERA5%3A+data+documentation#ERA5:datadocumentation-Instantaneousparameters). We have added this to the manuscript.

L137: Add a sentence explaining how the Luhar et al. parameterization is improved over Price and Rind.

Done.

[Figure]

L143: How do you re-grid the data onto the 2.5 x 2.5 degree grid?

We start with the coordinates of each lightning flash reported by ISS-LIS. We search in ERA5 reanalysis data the value of the vertical velocity in the position of the flash and up to 5 cells away (in latitude and longitude directions). Finally, we average the vertical velocity over all the selected cells (see Fig. C1). We have explained it in the manuscript.

L159-160: What was the correlation between updraught mass flux and the ratios?

The correlation is explained in the following paragraphs. We find a cuadratic correlation between the updraught mass flux and the ratios.

Figure C1: ERA5-grid cells that are used to average the vertical velocity for each lightning flash.

L168: Figure 3: The vertical lines used to show the binned data are confusing. Wouldn't it make more sense to show the values as x's

or +'s? The binned data still shows several ratios of 0 and 1. This suggests that the bins may be too small. How much would the results change by if you increased the bin size by a factor of 2-5?

We have removed the vertical lines.

We show in Fig. C2 the same figure as Figure 3 of the manuscript but increasing the bin size by a factor of 2 and 5. Comparison of the quadratic regression with Fig. 3 shows that there are not significant differences in the fitting of the ratio LCC(>9 ms)/typical lightning. On the contrary, the bin size has a larger influence in the ratio LCC(>18 ms)/typical lightning. As we stated in the manuscript, LCC(>18 ms)-lightning flashes are rare. This is also reflected in the influence of the bin size on the parameterization.

[Figure]

*Figure C2: Same as Fig. 3 of the manuscript but increasing the bin size by a factor of 2 (left column) and by 5 (right column).*

This sensitivity analysis has been added to the manuscript.

L172-179: What do you do for values LCC9 (LCC18) values above 0.5 (0.3) kg m-2 s-1?

We do not include those values in the quadratic and cubic fittings. We assume that the ratios are zero for fluxes above 0.5 (0.3) kg m-2 s-1. We have included this to the manuscript.

L185: I see "good" agreement for mass fluxes < 0.3 as opposed to 0.5. Are both quadratics used between 0 and 0.5 or is only the LCC9 one used for 0-0.5?

The quadratics are used between 0 and 0.5 kg m-2 s-1 in the case of LCC(>9 ms)-lightning and between 0 and 0.3 in the case of LCC(>18 ms)-lightning. We have included this to the manuscript.

L190: Is this the updraught velocity from EMAC? How does the updraught velocity from EMAC compare to that from ERA5?

Yes, the updraught velocity used in the simulations is calculated from EMAC. We show in Fig. 3C a comparison between the mean updraught mass flux at 440 hPa pressure level extracted from ERA5 and from a EMAC simulation. The values extracted from ERA5 correspond are monthly averaged using the 1-hourly product in a 0.25x0.25 degrees resolution grid for August 1999, while the values

extracted from EMAC are monthly averaged from a T42L90MA resolution simulation for August 1999. As we explain in the manuscript, the absolute value of the updraft depends on the grid resolution. However, Fig. 3C shows that there is a fairly good agreement in the spatial distribution of the updraft between ERA5 and EMAC.

[Figure]

*Figure 3C: Monthly averaged updraught mass flux at 440 hPa pressure level extracted from ERA5 (left) and from a EMAC simulation (right).*

L224: Why do you give values for two resolutions? How does the horizontal resolution vary between T42L90MA and T52L41DRL? Does T stand for triangular?

There was a typo. T52L41DLR should read T42L41DLR.

T42 stands for a triangular truncation at wave number 42 for the spectral core of ECHAM5 (see https://doi.org/10.1175/JCLI3830.1 for more details). L90MA stands for 90 vertical levels from the surface up into the middle atmosphere (MA), mid upper layer is 0.01 hPa (~80 km). L41DLR means 41 vertical levels from the surface up to approx. 10 hPa (mid of uppermost layer).

L235: When evaluating the schemes you may want to evaluate them separately over land and water. Parameterizations with separate land and ocean schemes should do much better at capturing the land/ocean contrast. You may want to indicate in Table 2, which these are.

Done.

L242-245: Be sure to emphasize the observed value of 2:1. It seems to get lost in all of the discussion of ratio shifting.

Done.

L268: You state that the lightning parameterization underestimates the ratio over the oceans; however, in Figure 7, I see larger values for Pcth+Aprec than for ISS-LIS, especially between 25 and 45S. What am I missing here?

The first column of Figure 7 shows the observed ratio between 2017 and 2020. These observations are provided by ISS-LIS, a Low Earth Orbit satellite. Therefore, areas with a low occurrence of thunderstorms (for example, areas over the ocean) are not well represented. This is the reason why we added Figures 9-12, where we show differences in the ratio only for areas with observations. Figures 9-12 clearly show that the ratio over the ocean tend to be higher for observations provided by ISS-LIS. We have added a remark after the statement "they tend to underestimate the ratio over the oceans ".

L265: Figure 9-12: Do not shown more than 2 significant digits for the correlation.

Done.

L265: Figure 9-12: In caption state whether positive values mean the parameterization has overestimated or underestimated the ratio. Ideally, positive values should indicate that the model has a high-bias.

Done. We have changed the colors accordingly.

L265: Figure 9-12: It is a bit overwhelming looking at 4 6 plot panels. You may want to show 4 panels instead (Perhaps remove the LCTH and Aupdr plots; the former because it is similar to the more widely used Pcth scheme and the latter because it performs poorly).

This is a technical paper that aims to serve as a guide to simulate LCC-lightning with the EMAC model. We think it is worth showing the results for all the lightning parameterizations included in EMAC so that the users can decide which is the most adequate for each particular case.

L265: Figure 9-12: From a reader's perspective, it might make more sense to show biases in LCC9 flashes as opposed to biases in the ratio

Biases in LCC-lightning are mostly influenced by biases in lightning (which can be seen in Fig. 4). Showing biases in the ratio is interesting because they indicate the isolated effect of the updraft in the simulated LCC-lightning distribution.

L285-286: The description of what is shown in Figure 13b is unclear. Please re-write.

Done.

L308: … but why is the impact of aerosols larger for LCC flashes than for "normal" flashes?

The answer to this question is not obvious and is out of the scope of this paper. Bitzer (2017) proposed that electrification processes in thunderstorms could play a role in the ratio of LCC-lightning to typical flashes. According to his hypothesis, a slower electrification could produce larger regions with charged particles before lightning is initiated, enabling the discharge to produce a LCC-lightning flash. Previous studies, such as Tao et al. (2012), have proposed that aerosols play a complex role in the electrification process. Therefore, we expect that aerosols can also play a role in the occurrence of LCC-lightning. However, what is the role of aerosols in the production of LCC-lightning is still unclear. More observation and micro-physical modeling efforts are needed to understand the possible relationship between aerosols and LCC-lightning.

In the discussion section of this manuscript, we observe that the LCC-lightning parameterization based ONLY on the updraft does not perform well in regions with high concentrations of aerosols. Therefore, we propose that aerosols could influence the occurrence of LCC-lightning. However, we are not able to propose a possible mechanism to explain the relationship between LCC-lightning and aerosols. We have added this to the manuscript.

L310-316: In DJF the ISS-LIS has few pixels with data over the North Atlantic while the model has high values there (see Figure 7a-b). Is some of the "missing" seasonality in the observations due to sampling constraints?

Yes, this is a possibility. The observed ratio is not statistically reliable if the total number of observed flashes is too low. We have added this reasoning to the manuscript.

L317-339: These listings of underestimations/overestimations are tedious unless accompanied by some analysis as to why the ratios are too low or too high. Either delete or add some analysis or even speculation. Also, I'm not sure the repeated Blakeslee et al. (2000) citations are needed for something so basic.

We have deleted these paragraphs.

L340-364: Again, this section is tedious and perhaps unnecessary.

We have deleted these paragraphs.

L365: Be sure to remind the reader as to why a LCC parameterization would be useful.

Done.

L373: Is the novel scheme competitive with the cloud-top-height schemes when parameterizing total flashes or is its use best limited to the parameterization of LCCs?

The LCC-lightning parameterization is competitive when using typical cloud-top-height parameterization, as shown in Fig. 13.

**Technical Corrections:**

L1-2: This type of flashes are ... Flashes of this type are

Done.

L6: Previous reports ... Reports

Done.

L28: Despite the evidences ... Despite evidence

Done.

L32: LSS are mostly assembled by ... LLS include

Done.

L35: may lack of a high ... may lack a high

Done.

L37: are not useful to provide ... provide little

Done.

L84: clearer shown ... clearly shown

Done.

L213: Double check 5 hours; 1, 3, or 6 are more typical.

We confirm we selected 5 hours.

L281: higher correlation ... highest correlation

Done.

L282: lower correlation ... lowest correlation

Done.

L283: higher ... highest

Done.

L294: Table 3 caption: approximately values ... values

Done.

L296: showed ... shown

Done.

L319: When referring to Figure 9 be clear as to whether you are referring to DJF or the entire year.

This paragraph has been removed.

L393. mesosphere . ... mesosphere.

Done.

**Reviewer 2**

General Comments

Pérez-Invernón et al. utilized the Lightning Imaging Sensor to develop different LongContinuing-Current (LCC) parameterizations and compared the simulations with observations. This new parameterization could benefit both LCC simulations and lightning NOx studies.

We thank the reviewer for these encouraging comments and for the time spent in the revision of this manuscript.

Specific Comments

1. L12: What is the meaning of typical lightning? It should be typical total lightning according to the main text. If I am wrong, please correct me.

The reviewer is right. We have now written typical total lightning instead of typical lightning.

2. L66-69: It is necessary to explain why do you only mention LCC (>9 ms) and LCC (>18 ms). How about the longer LCC?

We have now added a discussion on the duration of the LCC-lightning from comparison between optical and radio signal provided by Bitzer (2017) using the Huntsville Alabama Max Meter Array. As we explain now, LCC(>18 ms)-lightning can serve as a good proxy for lightning-ignited wildfires [McEachron and Hagenguth. (1942), Fuquay et al. (1967)]. We do not include parameterizations for longer LCC-lightning flashes because they are rare and the total number of measurements is not enough. The low occurrence of LCC(>18 ms)-lightning and longer lightning is suggested in the conclusion.

3. L77: If I understand correctly, it is the total lightning distribution which agrees with that derived in Blakeslee et al. (2020). Please rephrase the sentence to make it clear and connect with the peak flash density and land-ocean contrast mentioned later.

Yes, the total lightning distribution is the one that agrees with Blakeslee et al. (2020). We have rephrased.

4. L81-92: It would be interesting to see more discussion about the meaning of the ratios between LCC and total lightning flashes. While the introduction section has explained why the ocean has a larger ratio, how about the cause and meaning of the maximums over land? Are these also weak convection?

We have added more discussion about the higher ratios over the ocean and over continental areas that are not lightning hotspots.

5. L94-95: It seems the ratio of LCC (>18 ms) to total lightning usually exists over the cells with large ratio of LCC (> 9ms) to total lightning. This indicates that LCC (>18 ms) is the subset of LCC (>9 ms) and explains the smaller number of LCC (>18 ms).

We have added this to the manuscript.

6. L113: As discussed in [Romps, D. M. (2019)] (https://doi.org/10.1029/2019GL085748), "the mixed-phase cloud region is bounded by the 273-K isotherm (where ice can first form) and the 240-K isotherm (where liquid drops freeze homogeneously)". They use IFluxT, defined to be the convective ice flux on the 260-K isotherm, which lies within the mixed-phase regions of clouds and is close to the 440-mbar isobar in a modern-day tropical sounding. I know it is a large work to do the sensitivity test, could authors point the importance of isotherm?

We have pointed the importance of isotherm.

7. L127-L128: Shouldn't the tropospheric and middle atmosphere processes include the effects of anthropogenic emissions? maybe authors want to emphasize the meteorological atmosphere processes?

We would like not to extend the length of the manuscript. We prefer referring to Jöckel et al., (2010, 2016) to a deeper description about the meteorological atmosphere processes included in the model.

8. L143: How did the authors re-grid the updraft? It is better to use the maximum updraft in the grid according to the updraft references.

We have now added a detailed explanation on how we have re-grided the updraft.

9. L159: Does "the possible relationship" stands for the relationship between ratio and updraught mass flux?

Yes. We have added this to the manuscript.

10. L171: It seems there are few points when the updraught mass flux larger than 0.3. Could this affect the regression?

Yes, there are few points above 0.3 km m$^{-2}$ s$^{-1}$ that affect regression. Below ~0.3km m$^{-2}$ s$^{-1}$, the data suggest a proportional relationship between the ratio of LCC(>9 ms)-lightning to typical lightning and the updraft. However, the data above ~0.3km m$^{-2}$ s$^{-1}$ suggests the opposite. The obtained parameterization of LCC(>9 ms)-lightning to typical lightning follows a quadratic relationship instead of a linear relationship with the updraft because the data above ~0.3km m$^{-2}$ s$^{-1}$.

11. L185: Did the authors get some model grids with updraught mass flux larger than 0.5? If so, the uncertainty of LNOx will also be large. If the authors have written some other manuscript implementing this parameterization, please tell readers the limitation.

Please note that the implemented LCC-lightning parameterization does not play any role formación the $LNO_x$. In L185, LNOX stands for the MESSy submodel that includes the lightning parameterizations. The lightning injected $NO_x$ ($LNO_x$) is introduced in the LNOX submodel following the typical total lightning frequency.

The comparison between the updraught mass flux between the ERA5 and the model is discussed in Section 4.1.

12. L235-L236: As mentioned in Luhar et al. (2021), their marine parameterisation yields flash rates that are approximately an order of magnitude smaller than the PR92. Why the authors get the larger difference (5:1 and 1:1)? Is it caused by the model resolution?

The obtained large difference over ocean by using $P_{cth}$ and $L_{cth}$ can be due to several factors. The model resolution is one of them. However, literature has shown that lightning parameterizations do not perform equally in different models [e. g., Tost etl a. (2007), Gordillo-Vázquez et al. (2019), …]. The reason is that each atmospheric model includes different parameterizations for atmospheric processes, such as convection. Luhar et al. (2021) used the ACCESS-UKCA model, while we have used the EMAC model. Establishing the reasons why the parameterization proposed by Luhar et al. (2021) performs differently in ACCESS-UKCA and in EMAC would require a deep comparison between both models, which is beyond the scope of this paper.

13. L268: Are "maximum values" compared with both land and ocean data?

No. In this first parameterization of LCC-lightning we do not include differences between land and ocean. However, it would be something interesting to be done in the future.

14. L282-L284: Interesting results of seasonal correlation. Do authors have any explanations?

We have proposed that the seasonal correlation could be due to the high influence of the seasons with high lightning activity for the development of the parameterization. The larger the lightning activity during a given season, the more observations included in the development of the parameterization during that season.

15. L315-L316: Are extreme values of updraft from reanalysis or simulation? If they only exists in one dataset, that may explain the bad agreement.

We have removed this paragraph following the comments from reviewer 1.

16. In the Discussion section, authors usually use "good agreement", "higher", "lower" to explain the figures. They need to come up with a better way like putting some figures in the Supplements and add use some indexes to judge "good or bad" and "higher or lower".

We have included the indicative values of "high", "medium" and "low".

Technical Corrections

1. L11: to find a global parameterization → to develop a global parameterization

Done

2. L13: It is better to give the full name of EMAC in the Abstract

Done

3. L31, L37: LSS are→LLS is

Done

4. L42: Add a space between degree symbol and N(S)

Done

5. L47: the process of separation of electrical charges → the process of electric charge separation

Done

6. L51: could be helpful to ..... and (to) .…

Done

7. L63: lightning with ... and (with) …

Done

8. L71: The TRMM-LIS ended in 2015 and a similar instrument onboard the ISS replaced it for a 4 years mission …

Done

9. L76, L81, L93: It would be better to use a,b,c instead of first/second panel to point out the subplot. Please also check other figures.

Done

10. L82: are also regions with →coincide with

Done

11. L84: is clearer shown ... showing the ratio of →is more clearly shown by the ratio of .... in Fig 1c.

Done

12. L89: between 35◦ N and 35◦ S latitude →between 35◦ N and 35◦ S

Done

13. L90: All these regions are well-known regions for →All these regions are wellknown for

Done

14. L120: "section" →"Section"

Done

15. L129: "by using" →"by"

Done

16. L134: "a novel combination .... suggested by us" →"our novel combination"

Done

17. L138: What is the definition of the "scaling factor" in Table 1? Because I do not see the same numbers in these references, authors may have their own definition.

The scaling factor "ensures a global lightning occurrence rate of ~45 flashes per second". This is different for each model, resolution, etc… . Several changes have been introduced to EMAC since Tost et al. (2007). Therefore, we have calculated the scaling factor for the employed resolution and the current version of EMAC. We have now mentioned this in the masnucript.

18. L142: "prepare" → "process"

Done

19. L143: The sentence can be simplified into "the global 1-hourly averaged values of the vertical velocity at the 450 hPa level between March 2017 and March 2018 are re-gridded onto a 2.50◦ × 2.50◦ latitude and longitude grid, which it is similar to that typically used in global chemistry climate models."

Done

20. L197-L199: This is duplicated with L195-L196. If I misunderstand, please correct me.

Thank you. We have rephrased.

21. L206: "with a quadratic Gaussian grid of 2.8◦ × 2.8◦ in latitude and longitude" → "with a .8◦ × 2.8◦ quadratic Gaussian grid"

Done

22. L216: Please rephrase the sentence. The grammar is wrong.

Done

23. L231-L232: "figure 4" → "Fig. 4"

Done

24. L263: "Figure 7 and 8" → "Fig 7. and 8."

Done

25. L264-L265: "Figures 9-12" → "Fig. 9-12"

Done

26. L276: "off" → "of"

Done

27. L300-L301: Please add the references.

Done

28. L305: Which ratio?

Ratio LCC(>9 ms) to all lightning added

29. L310: "suggests" → "suggest"

Done

30. L317: "season" → "seasons"

We have removed this paragraph following the comments from reviewer 1.

31. L319: "parameterization" → "parameterizations"

We have removed this paragraph following the comments from reviewer 1.

32. L342: "entail" → "entails"

Done

33. L375: "entail" → "entails"

Done

34. L386: "that will serve to complement" → "that can complement"

Done

35. L386: "will serve to improve the" → "will improve"

Done

36. L387: "will serve to improve" → "is needed to improve"

Done

37. L391: Too many "serve to" are used in the Conclusion. Please polish it.

Done

38. Fig 13. "Each point represents a different season." It is better to use different opacity (or other symbols) for each point. Otherwise, readers do not know which point is which season.

We have included the season for each point

**Reviewer 3**

General Comments

We thank the reviewer for the time spent in the revision of this manuscript.

My primary concern regarding this manuscript is the scope or the purpose of the LCC parameterization. As the authors described, the main reason to perform this study is for the prediction of wildfires ignited by LCC. In that sense, the analysis of LCC over ocean doesn't make much sense.

Please note that, as we mention in the manuscript, the developed LCC-lightning parameterization can also be used to parameterize sprites. Sprites can also occur over the ocean [e. g., Pasko et al. (2012), Gordillo-Vázquez and Pérez-Invernón (2021)]. We have added some more discussion about the role of LCC-lightning in the occurrence of sprites.

However, it may help enhance the understanding of land vs ocean differences in convection frequence and intensity, and in turn to help more accurate lightning NOx implementation in atmospheric modeling in the future. I suggest that the authors provide some discussions in that direction. Even though majority of the lightning NOx models at this time don't distinguish the flash types due to limited observations and knowledge regarding lightning NOx production efficiency, it

may not be the case in the future with more observations and studies becoming available. The manuscript is in general well written and clearly presented.

We thank the reviewer for these encouraging comments. We have implemented some more discussion on the differences between land and ocean in the distribution of LCC-lightning. As the reviewer has pointed out, the current LCC-lightning parameterization does not play any role in the lightning-produced $LNO_x$. However, future parameterizations of $LNO_x$ could introduce lightning types.

Specific Comments

Since the terms LCC (>9 ms) and LCC (>18 ms) are used repeatedly through the manuscript, it would be good to define it as LLC9 and LCC18 from the first appearance, and that will simplify the descriptions and even easy to comprehend.

We thank the reviewer for this suggestion. However, we have already used this notation in other manuscripts. We prefer to maintain this notation for coherence.

Line 25, how LCC-lightning relates to positive lightning?

As we mention in the introduction, the VLF of LLS cannot detect the continuing current if the flashes are not in their vicinity. Therefore, a global analysis of the polarity of LCC-lightning is not possible. However, Bitzer (2017) analyzed the radio signal emitted by LCC-lightning flashes in the US. Bitzer (2017) found that 55.2% of the detected LCC-lightning flashes were positive. We have added this to the manuscript.

Line 28, "Despite the evidence of the role of LCC-lightning in lightning-ignited fires", any reference or observations?

The references including these evidences are mentioned some lines before in the same paragraph.

Lines 284-285, "On the contrary, the higher correlation coefficient over ocean is reached during March, April, and May". Does this mean that the maximum lightning activity over ocean occurs during these months?

The discussions regarding the correlation coefficients on Page 10 (Figure 13) are rather vague due to the low values of the correlation coefficients and the little differences among the different parameterizations, especially for those over the ocean. It could enhance the credibility by providing the significance values (p-value) and the confidential intervals of these numbers.

Throughout the seasons of the year, the lightning contribution from oceanic regions remains relatively constant [Blakeslee et al. (2014): **https://doi.org/10.1016/j.atmosres.2012.09.023].** We have changed this phrase and added this reference. We have also extended the discussion on the seasonality of the correlation coefficients and added the seasons to the lower pannel of Fig. 13.

Lines 360-363, do other years show similar seasonality over different regions?

Yes. We have included this in the manuscript.

---

## Referee Report (RR1)

**Major Comments:**

Author has responded to each of the reviewers' comments.

In the conclusion section be sure to state how the spatial distributions of the LLC(>9ms) and LLC(>18ms) distributions differ from the spatial distribution of total lightning.

**Minor Comments:**

I would also prefer LLC9 and LLC18 as acronyms.

L196: It is a bit confusing when you state that the ratio is assumed to be zero for fluxes greater than 0.5 (0.3). → State that the ratio is unchanged and equal to 0 for fluxes greater than 0.5 (0.3).

L241: Output every 5 hours is unusual because 24 /5 is not an integer.   Do you mean 3 or 6?

L379: How many more years of data are needed before robust conclusions can be obtained for LCC (>18 ms) flashes can be obtained?

Figure 1 caption: Explain why the ratio plots are undefined over a portion of the globe.

Figure 7 caption: Seasonal observed (top panel) and simulated (rest of panels) ratio → Seasonal observed (left panels) and simulated (right panels) ratio

Figure 8 caption: See correction for Figure 7 caption.

**Grammatical Suggestions:**

L22: during more than → for more than

L58: Why do you use a different font for LNOX?

L65: and in the range of latitudes → for latitudes

L80: replaced it for a 4 years mission starting in March 2017 … → replaced it in March 2017 and is still sampling latitudes between 54.3 N and 54.3S (Blakeslee et al., 2020) as of January 2022

L85: What does typical mean? Is this the ratio of LCC/(All_Flashes) or LCC/(All_Flashes – LCC)? I'm guessing the latter but am uncertain.

L91: activity coincides → activity coincide

L92: distributions more → distributions is more

L99-100: high production of LCC(>9 ms)-lightning respect to all lightning → high ratio of LCC (> 9 ms) to total lightning.

L106: nearly similar → similar

L108: indicates that → is not surprising as

L108: the subset → a subset

L110: obtained total → total

L160 → lightning parameterizations and scaling factors used in this study are summarized in Table 1.

L165: combination → combining

L268: choice lightning → choice of lightning

L303: LIS missing → LIS to miss

L327: spatial distribution → spatial distributions

L328: comparing with observation → comparing with observations

---

## Author Response (AR2)

We thank the reviewers for the revision of this manuscript. In this document, we provide a detailed answer to all the comments raised by the reviewers (blue).

**Reviewer 2**

**Major comments:**

Author has responded to each of the reviewers' comments.

In the conclusion section be sure to state how the spatial distributions of the LLC(>9ms) and LLC(>18ms) distributions differ from the spatial distribution of total lightning.

Done.

**Minor comments:**

I would also prefer LLC9 and LLC18 as acronyms.

We prefer maintaining the notation LCC(>9 ms) and LCC(>18 ms) for consistency with other works.

L196: It is a bit confusing when you state that the ratio is assumed to be zero for fluxes greater than 0.5 (0.3). → State that the ratio is unchanged and equal to 0 for fluxes greater than 0.5 (0.3).

Done.

L241: Output every 5 hours is unusual because 24 /5 is not an integer. Do you mean 3 or 6?

Indeed, we usually choose 5-hourly output because 24/5 is not and integer in order to avoid systematic local biases w.r.t. to the solar zenith angle. In other words, with 5 hourly data, we get at any position of the globe also information about the diurnal cycle. Output every 12,6, or 3 hours will - at a given geographic location – always be for the same local solar time (and zenith angle).

L379: How many more years of data are needed before robust conclusions can be obtained for LCC (>18 ms) flashes can be obtained?

The total number of LCC(>9 ms) and LCC(>18 ms) flashes observed during one year are $2.3 \times 10^5$ and $2.6 \times 10^4$, respectively. Therefore, we estimate that 10 years of data are necessary to obtain robust conclusions for LCC(>18 ms)-lightning flashes. We have added this to the manuscript.

Figure 1 caption: Explain why the ratio plots are undefined over a portion of the globe.

Done.

Figure 7 caption: Seasonal observed (top panel) and simulated (rest of panels) ratio → Seasonal observed (left panels) and simulated (right panels) ratio

Done.

Figure 8 caption: See correction for Figure 7 caption.

Done.

**Grammatical Suggestions:**

L22: during more than → for more than

Done.

L58: Why do you use a different font for LNOX?

We are now using the same font as for the rest of the text.

L65: and in the range of latitudes → for latitudes

Done.

L80: replaced it for a 4 years mission starting in March 2017 ... → replaced it in March 2017 and is still sampling latitudes between 54.3 N and 54.3S (Blakeslee et al., 2020) as of January 2022

Done.

L85: What does typical mean? Is this the ratio of LCC/(All_Flashes) or LCC/(All_Flashes – LCC)? I'm guessing the latter but am uncertain.

This is the ratio LCC/(All flashes). We have removed "typical".

L91: activity coincides → activity coincide

Done.

L92: distributions more → distributions is more

Done.

L99-100: high production of LCC(>9 ms)-lightning respect to all lightning → high ratio of LCC (> 9 ms) to total lightning.

Done.

L106: nearly similar → similar

Done.

L108: indicates that → is not surprising as

Done.

L108: the subset → a subset

Done.

L110: obtained total → total

Done.

L160 → lightning parameterizations and scaling factors used in this study are summarized in Table 1.

Done.

L165: combination → combining

Done.

L268: choice lightning → choice of lightning

Done.

L303: LIS missing → LIS to miss

Done.

L327: spatial distribution → spatial distributions

Done.

L328: comparing with observation → comparing with observations

Done.

**Reviewer 3**

The Sect. "Discussion" still needs some revisions.

1. The authors wrote "... analyze the seasonal and spatial distribution ..." at the beginning, but they discuss the spatial distribution first, and then the seasonal one. It is better to modify the sentence to match them.

Done.

2. The logic of the discussion is not clear because of the deleted parts. The authors should take care of it. For example, the spatial distribution is discussed for both land and ocean. But the seasonal one just focuses on the ocean. I suppose they can split it into two subsections and make the discussion clearer to readers.

Done.